# Towards Robot Vision Module Development with Experiential Robot Learning

## Abstract

In this paper we present a thrust in three directions of visual development using supervised and semi-supervised techniques. The first is an implementation of semi-supervised object detection and recognition using the principles of Soft Attention and Generative Adversarial Networks (GANs). The second and the third are supervised networks that learn basic concepts of spatial locality and quantity respectively using Convolutional Neural Networks (CNNs). The three thrusts together are based on the approach of Experiential Robot Learning, introduced in previous publication. While the results are unripe for implementation, we believe they constitute a stepping stone towards autonomous development of robotic visual modules.

## 1 Introduction

Can a robot learn from its environment? Can a robot learn to appreciate multiplicty, i.e. the concept of plurality of instances of objects? Can a robot have attention mechanisms that allow it to focus on a particular object in its visual field? Can a robot somewhat understand and communicate the concept of an object's location natively?

We attempt to address those questions, particularly as pertains to a child robot that is open-ended and designed without a strict purpose. Our investigations stem from the motivation of creating autonomous child robots that learn in-the-wild, on-the-fly. We are not attempting to design elaborate or tailored algorithms to solve a particular problem. Instead, we attempt to tackle these difficult challenges with the simplest of solutions.

Hand-engineering solutions can be, while necessary in some cases, a limiting factor in deriving solutions for autonomous development. As such, we want to equip our open-ended child robots with the capabilities to learn. We thus present some weakly-supervised and semi-supervised models, in attempt to share the insights we generated by runnning these experiments.

## 2 Background

The advent of Deep Learning has brought about a revolution in Perception systems and the way they are designed. Deep Neural Networks are now used for a wide range of applications from Machine Translation Singh et al. (2017) to Photo-editing (Lee et al., 2017). We noticed however that most of the techniques rely on supervised methods to realize and achieve learning coupled with the Stochastic Gradient Descent (SGD) algorithm, such as (Liu et al., 2015), (Redmon et al., 2015), (Girshick, 2015) and (Ren et al., 2015). Since the amount of annotated data is but a subset of the amount of data available in general, we are motivated and inclined to develop systems that leverage this body. In addition, data acquired on-the-fly, i.e. in real-time, are also usually unannotated.

Consider the example of images. While a network may be trained on a million labelled images of different objects, those images are but a subset of the total amount of images currently available, and potentially acquired in the future. In addition, consider the amount of effort required to annotate incoming images from a video feed with bounding boxes, if at all possible, so that an Object Detector can learn to detect new objects, including the prospect of taking out a deployed, production system for re-training. This is particularly of import for autonomous robots with local cognitive development, i.e. over-the-air updates are ineffective.

## 2.1 Experiential Robot Learning

It is for those reasons outlined above, among many others, that we introduced the method of Experiential Robot Learning (ERL) in (Aly & Dugan, 2017). It is proposed to propagate the principles of Self-Improvement and Scalability. Briefly, ERL sets the stage for the development of autonomous agents capable of propelling themselves in terms of skill and efficacy. We want robots to learn in-the-wild, from experiencing the real world. We do believe this will require a cognitive bootstrap. In our previous work (Aly & Dugan, 2017), we demonstrated an implementation that follows ERL by re-purposing the fine-tuning technique for CNNs. This allowed the robot to create an expanding vocabulary of visually-recognizable objects, with the aid of a human teacher.

We don't have a qualm with the idea of using supervision, or supervised learning. However, we do recognize how it can be a limiting factor to self-improvement and scalability. Furthermore, in some cases, supervision is not feasible such as the case of annotating an image with bounding boxes.

Therefore we make the distinction of accessible and inaccessible supervision. In the case of our previous demonstration we see it as accessible supervision or annotation. Specifically, the supervision is provided in the form of a pronounced label, to be acquired by the robot through voice recognition; The human speaks the label to the robot, in a conversational manner.

The rest of this paper is organized as follows. In the coming section, Implementation, we will elaborate on the three thrusts we put forth towards the development of robot visual modules with ERL. Following this we present a survey of related work. In section 5 the results of our experiments are presented with a discussion. Finally, concluding remarks are given in section 6.

## 3 Implementation

In this section we will elaborate on the implementation details of the proposed systems. All the networks are coded in TensorFlow (Abadi et al., 2015) using the Keras API (Chollet et al., 2015). The computations are performed locally on a desktop computer with an Nvidia Titan Xp GPU.

Ideally, we want the models and the computations to be deployed locally on the robotic agent. However, the computational demands of Deep Learning do not facilitate this currently. In the future, upon further optimization of our code and models, as well as the advancement of computational platforms in terms of hardware and software, it should be possible. In addition, the scope of this paper does not extend to a physical robotic implementation because the models themselves while insightful are yet unripe.

### 3.1 Semi-Supervised Object Detection

The aim of this module is to provide a self-guiding mechanism for the purpose of object detection. It follows the principles of Soft Attention and the general architecture of Generative Adversarial Networks (Goodfellow et al., 2014). The idea is to have the model correct itself on its object proposal, using a CNN trained on ImageNet as the Discriminator.

In essence, we transformed or reframed the Object Detection problem into a something of a regression problem. Initially, we wanted to follow the Hard Attention path by regressing on Bounding Box coordinates. However, our attempts were unsuccessful mostly due to the requirements of differentiability for Deep Learning models.

The model is separated into three stages as can be seen in Fig 1.

### 3.1.1 Stage One

The first stage of our model is the standard feature extraction. We noticed when we inspect the activation maps from a pre-trained CNN, on ImageNet, that they 'mask' out some features in the given image. Usually, those features also define the outline of the object, in essence masking it out. For examples, refer to Fig. 2.

The input to this stage is the raw image, and the output is the activation maps derived from a certain layer within the model. The choice of which layer's activation maps to get affects the overall

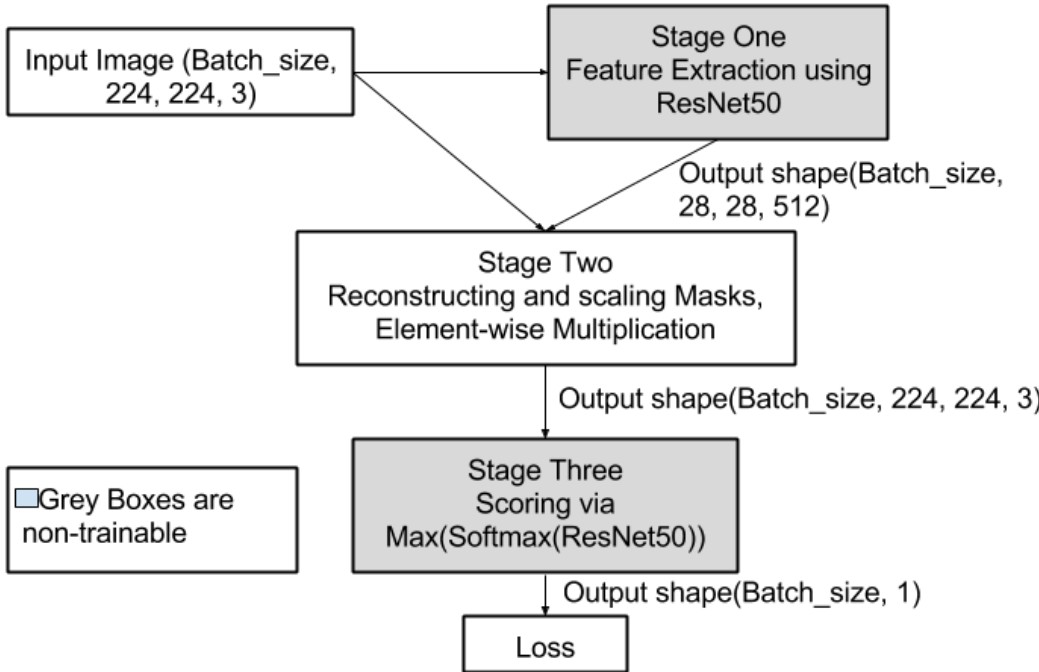

Figure 1: The diagram of the 3-stage object detector. The input image is passed to the First stage for feature extraction. The second stage takes the features as input and reconstructs and upscales them to the size of the image using Deconvolution layers to form masks. The input image is then element-wise multiplied with the masks, and passed on to the Third stage as a new input image. The Third stage is just a standard CNN with the softmax output. The max value of the softmax function is used as a 'score'. The closer the score is to 1 the better becuase it reflects a higher confidence in the soft attention mechanism

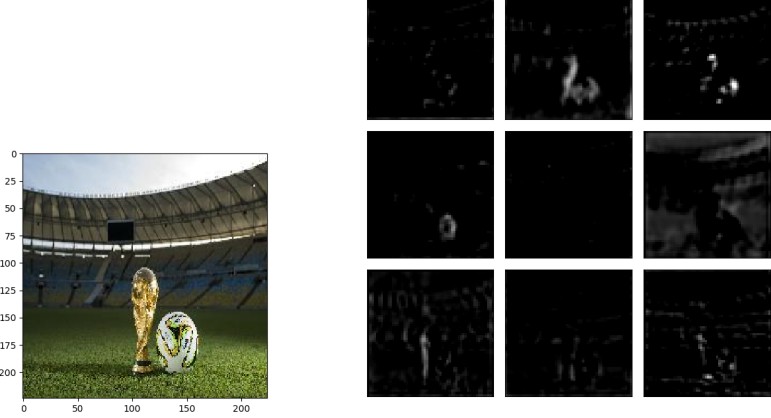

(a) An example input image featuring a soc- cer ball

(b) A sample of the activation maps pro- duced by a VGG16 network pre-trained on ImageNet

Figure 2: The figure shows the original input and the respective activation maps. the masking effect described can be observed, especially in the middle left map

performance of the model. That is, if for example we retrieve the maps of one of the introductory layers then we will skip the potential of entirely masking out the object of interest. That is because

the introductory layers in CNNs detect features like edges and curves. Similarly, if we choose the deepest layers' activation maps, then the maps may be too abstract to mask out desired object.

Note that as we travel down the CNN, from an information theoretic perspective, the features become more abstract and less correlated with the spatial dimensions of the input. Similarly, in the introductory layers of the CNN, the features are less abstract and highly correlated with the spatial dimension of the input. While this is fine for Object Recognition, we want a balance between Spatial Correlation and Abstraction.

Also, ideally, we want to use as large an input as possible. This is because much information is lost due to the pooling operations in CNNs, and we need to recover this information during the next stage.

### 3.1.2 STAGE TWO

This is the most important and critical stage in this model. This stage defines how well the model may perform for the given task. However, it is also constrained by the requirement of differentiability. In this stage we acquire activation maps from Stage One as input. The next step is to use that information in a meaningful way to our task.

The next step is to attempt to recover information and rescale those maps. This is done through the operation of Deconvolution, or Upsampling. Deconvolution, however, has the potential to, not recover, but add and reorganize information through training. For that reason we prefer it to Upsampling.

The feature maps are thus rescaled and reconstructed to match the size of the original input image. We then attempt to use those adjusted maps as masks to contrast the object in question to its background. This is how we use the soft attention mechanism. The feature matrices are multiplied directly, element-wise, with the input image matrix. In this way we apply the mask directly on the input image.

As mentioned earlier, we initially wanted to input the features as is and concatenate it with some random seed into a Bounding Box Proposal network. This is the Hard Attention path. The proposed Bounding Boxes, vectors of 4 integers, would then be regressed upon. Regression in that case was attempted by first cropping the image according to the proposed BBox dimensions and attempting to 'score' the crop.

The scoring function is possible by passing the cropped image through another CNN and checking the maximum element of the output softmax function. However this approach did not produce results for us. Therefore, we present here a different approach. Yet we saw fit to mention this technique as well in hope it kindles inspiration within the community.

### 3.1.3 STAGE THREE

The third stage is comprised of a standard CNN pre-trained on ImageNet. This stage in effect, acts as the discriminator if we follow the GAN architecture. It is static and non-trainable. The output of it is a 1000-wide matrix representing the 1000 classes of ImageNet dataset. Note that this discriminator can be replaced with other CNNs that are tailored to classify a specific type of object(s).

The CNN accepts as input the output of the second stage. Since it is agnostic, it treats it as a normal incoming image. A MAX function is applied to the output of the CNN, to obtain what we call the Score. The score represents the confidence of the CNN that the given image belongs to a certain class.

The idea here is to use the CNN to evaluate our soft attention mechanism. If the mask reconstruction process was solid, it would yield a high score since the passed image will represent clearly an object belonging to one of the 1000 ImageNet classes.

Finally, we regress on the score by providing a static target of '1'. This means that the model will constantly strive to reach the target, and never completely succeed given the nature of the CNN softmax function.

## 3.2 Object Location Network

We want our robot to be able to qualitatievly judge an object's location in its field of view. This is in contrast with other deterministic approaches, such as using bounding boxes for example. For our purposes of self-development in open-ended systems, location accuracy is not of critical import. An example application is asking the robot to navigate in the general direction of an object such as a trash bin.

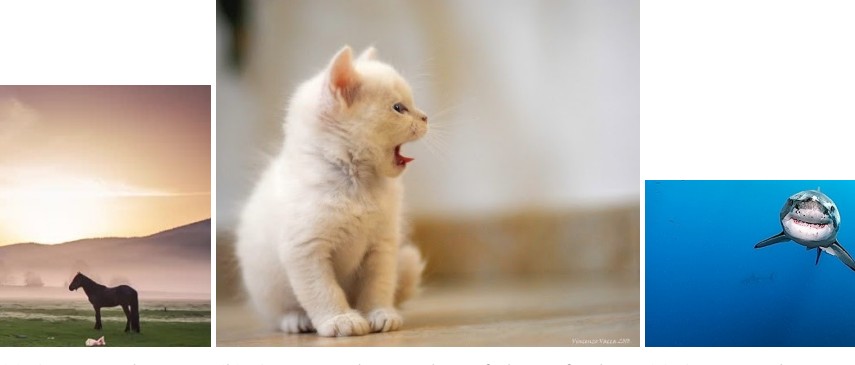

(a) An example member of the Bottom Center class
(b) An example member of the Left class, note here that the object occupies both the Top and Bottom Left of the image
(c) An example member of the Top Right class

Figure 3: The figure shows members of the different classes. In total there are 9 classes an object can belong to in our Object Location classifier

In this light, we repurposed conventional CNN classifiers to fit this need. We did so by again leveraging the masking behavior of activation maps but in a different way. We created 9 labels, namely Left, Right, Center, Top Left, Top Right, Top Center, Bottom Left, Bottom Right and Bottom Center.

We then compiled a custom dataset featuring objects exclusively in only the outlined locations, with an otherwise plain background. See Fig. 3 for examples of members. The idea here is that the classifier can encode the spatial location of the activation.

Fig. 4 outlines the architecture of the Location CNN. The intuition behind building the network in this way was as follows. We already knew and described the masking behavior of the activation maps of regular CNN classifiers. We wanted to create relatively specialized large filters to act as 'pass gates', or in another sense encoders, for the activations. We provide the parameters in the diagram as there is no need for much tuning, as will be demonstrated in the Results and Discussion section of the paper.

## 3.3 Plurality Autoencoder Network

This network represents an effort to further understand and tackle the concept of multiplicity. Plainly put, multiplicity refers to the possibility that numerous object instances can exist. Again, we are taking the perspective of a child robot that knows very little about our world. The ability to visually recognize multiplicty or plurality means an agent can start the task of learning to count.

We adopted the autoencoder paradigm for this task in attempt to understand better the idea of how a CNN handles plurality. At first, however, we started with a modified VGG16 architecture and tried to form the problem into a classification problem. That is, if an image had two object instances, label it as "TWO", and so on.

We did not obtain promising results from this approach as our validation accuracy saturated around 0.6. We, of course, could have driven it lower by engineering a network to solve the classification problem. However, that was not our goal from the start. We wanted to see if CNNs are capable of ingesting the concept of plurality natively.

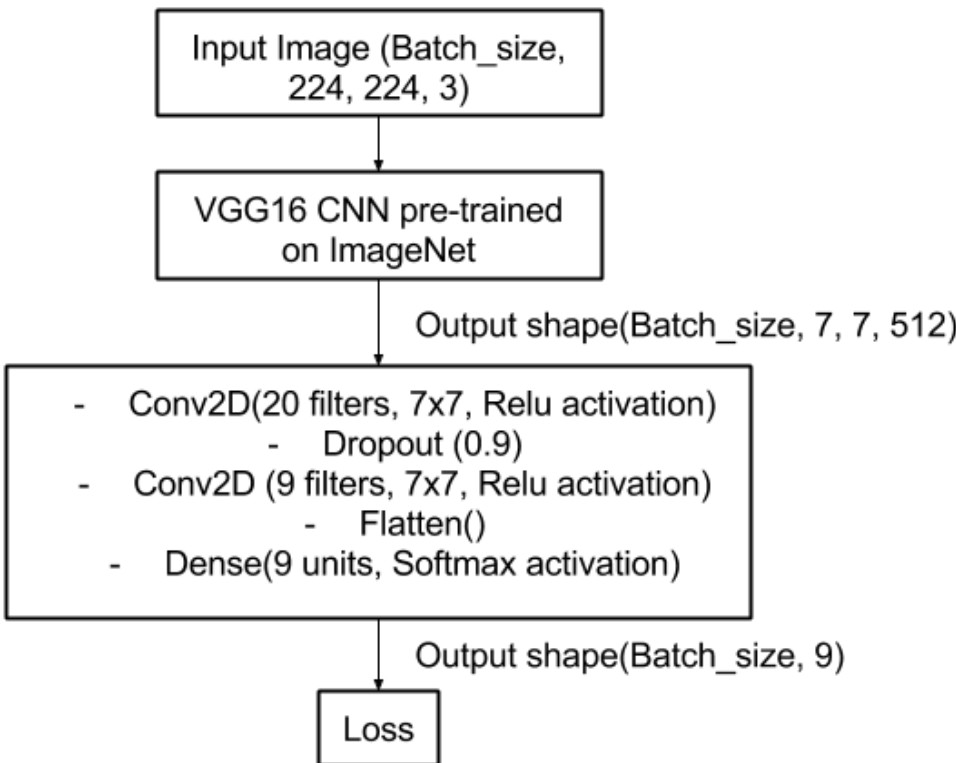

Figure 4: A diagram of the Location Network. Essentially, it is a VGG16 CNN feature extractor followed by a few convolutional layers of somewhat large filters. The large filters constitute the basis of our masking scheme

Therefore, we worked on the autoencoder architecture described here. The rationale was simple: we wanted to understand the problem more fundamentally. Was the CNN was suffering from a mapping problem, i.e. mapping the instances to the correct labels in its high-dimensional space? Or was the information of object plurality get lost, if it was picked up at all in the first place?

A dataset was compiled of more than 800 images, over 5 classes from "One" to "Five". The images feature the corresponding number of objects, whether homogenous or heterogeneous. The diagram of the autoencoder model is given in Fig. 5

## 4 RELATED WORKS

Our work criss-crosses many fields though in itself is relatively considered a niche. We will start our discussion by presenting work from the area of Developmental Robotics. First comes our main source of inspiration, the work in (Sigaud & Droniou, 2016). It served as the plateau for the development and synthesis of our research efforts, despite the work itself being without an implementation. Also, Nuovo et al. (2014) addressed grounding numbers and counting in child robots developmentally. However, their approach does not propose counting visually using deep CNN models.

In the vast area of Computer Vision, there is a plethora of work on the topic of Object Detection. For example, (Ren et al., 2015) and (Liu et al., 2015). Those approaches, however, rely upon the availability of annotated data for training. Even if we accepted this step, as we did with training our CNNs on ImageNet, we could not hope to improve upon them in the future. This is because future improvement, i.e. retraining, would require also annotations for the new data.

There are approaches for unsupervised object detection, of course. For example, the system in Siméoni et al. (2017) uses a similar approach to ours. Though instead of using regression as we do,

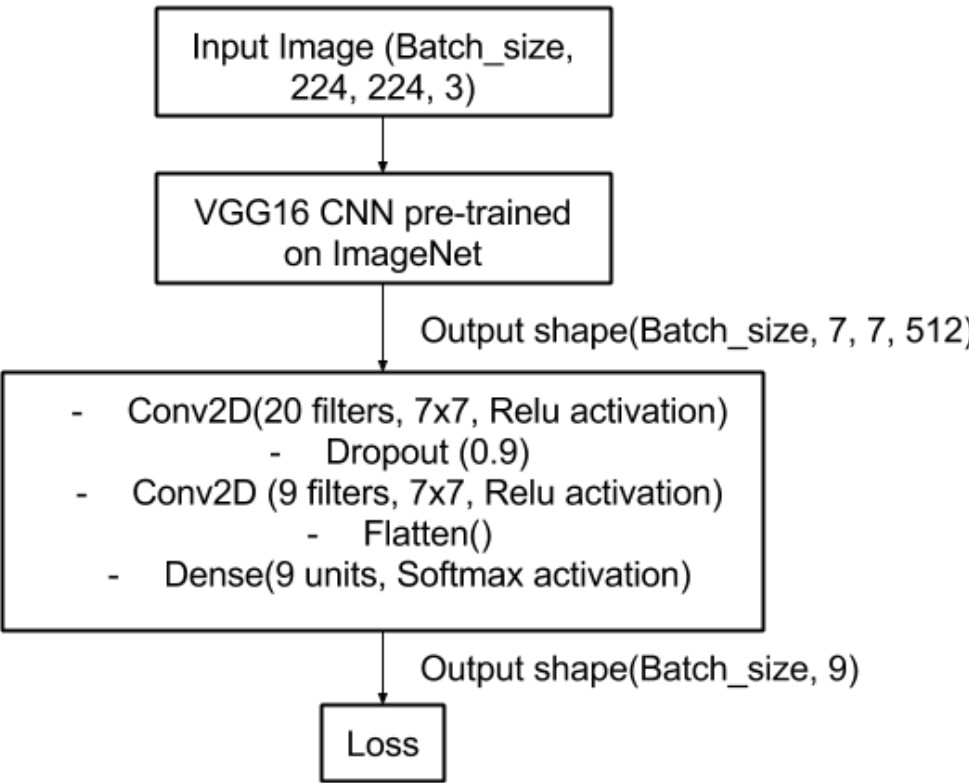

Figure 5: A diagram of the Plurality Autoencoder Network. The encoder is a ResNet50 CNN pre-trained on ImageNet. The last activation layer is taken as the final encoding. The encoded image is then deconvolved over 6 stages to reconstruct the image. Loss is calculated as a factor of the original image

they use the CNN activation maps to estimate object saliency. Furthermore, researchers in Wei et al. (2017) tackle unsupervised object detection using co-localization techniques. Their focus was on leveraging entire datasets of correlated images instead of operating on a single image.

In the area of counting using CNNs, there is also a formidable body of work we can draw from. For example, Segu et al. (2015) attempts to use Transfer Learning as means to address the problems of counting and also localization of objects of interests. Their work is tested on MNIST and a synthetic Pedestrian dataset. In addition, Cohen et al. (2017) uses an Inception-like architecture against subsets of the input image, i.e. receptive fields, to detect the count of objects. Finally, yet another example is the work in Lempitsky & Zisserman (2010). They use a metric called MESA (maximum excess over subarrays), to estimate the distance between the predicted counts and the target counts.

## 5 RESULTS AND DISCUSSION

In this section we present the results of applying our models to different in-house compiled datasets. We use our own datasets due to the limited computation resources available to us, in addition to the desire of fast prototyping.

Furthermore, the use of non-standard datasets is permissible since we are not claiming to advance the state-of-the-art on any of the current Computer Vision fronts. We are only presenting alternative approaches that yield some promise, along with the sharing our insights.

## 5.1 SEMI-SUPERVISED OBJECT DETECTION

In our experiments, we use a batch size of 64, Mean Absolute Error as the loss function and the Adadelta optimizer. We used a custom dataset of less than 2000 images. We found that the number of images does not affect the performance of the system. In addition, we also find that the number of epochs required to drive the loss down varies as the stage two configuration varies.

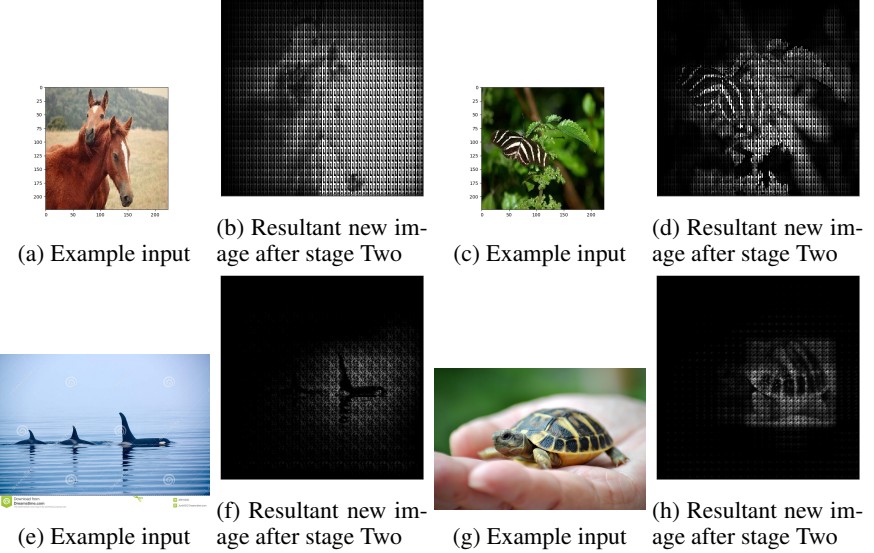

(a) Example input

(b) Resultant new image after stage Two

(c) Example input

(d) Resultant new image after stage Two

(e) Example input

(f) Resultant new image after stage Two

(g) Example input

(h) Resultant new image after stage Two

Figure 6: The figure shows the results of the Soft Attention mechanism. The results are of different configurations of Stage Two. It can be seen that correct masking is observable as most of the outer edges are ignored. However, still some of the background is present in the picture

From Fig. 6 it can be seen that there is some selection or focus given to the object of most interest in the input image. We provided different examples to highlight the different configurations and their results.

Plainly, by different configurations we refer to the virtually unbounded variation in hyper-parameters of the Deconvolution layers. Specifically, we vary the kernel size, and the number of filters in each stage. The addition of Dropout or BatchNormalization layers is also a factor. Furthermore, we can follow the Deconv layers with Convolution layers of a single stride. Those can further refine the constructed masks.

We can therefore see that in some cases, such as (b) and (d) attention is less refined than in (f) and (h). We can conclude that eventhough some models perform better than other, a more thorough analysis is necessary to characterize the behavior of different stage two models.

## 5.2 OBJECT LOCATION NETWORK

We used batch size of 60, the Categorical Crossentropy loss and the Adadelta optimizer. The system converges in less than 50 epochs to around 0.96 and 0.9 accuracy on the training and validation sets respectively. The validation set amounted to around 300 images.

The dataset used was comprised of more than 1500 images, spread over 9 classes. The images were of varying objects, though always singular and against a plain background.

From Fig. 7 The figure shows the training performance of the network. It tells us that this approach, though yet limited and constrained is no less promising. The validation accuracy is around 0.9 trailing a little the 0.96 training accuracy

In the future we would like to deploy this network on a robot. We envision it being used as means to have the robot steer itself by constantly keeping an object in the "Center" label. In addition, we intend to make the network more robust to noises by using more data with noise.

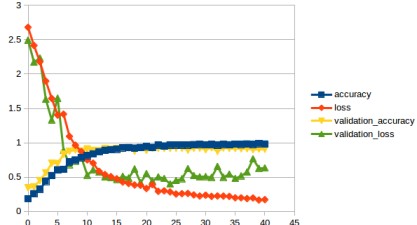

Figure 7: Training behavior graphs of the Object Location net

### 5.3 PLURALITY AUTOENCODER NETWORK

We ran our model with Binary Crossentropy loss, with the Adadelta optimizer and 80 epochs. The network is relatively slow to learn and converge, taking more than 50 epochs to start noticing learnings transfer to the validation set. Eventually, however, the network saturates at a certain loss. The training behavior is shown in Fig. 8.

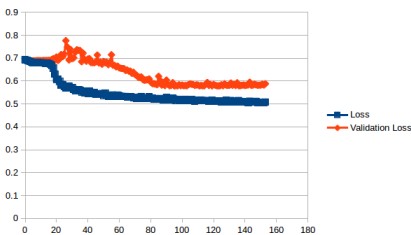

Figure 8: Training behavior graphs of the Plurality Autoencoder net

Despite the relatively high loss, upon checking the results over sample inputs as shown in Fig. 9, we noticed the network does understand multiplicity. For example we can see that the network produces 5 spoon-like shapes for an input image of 5 spoons. What this tells us is that the network has encoded information about the number of interesting objects in the image, and was able to successfully recover that information. This is further verified by checking that an image of a single object, such as the book shown in the figure, is reconstructed also as a single object.

Of course, with further engineering the loss could be brought down further. However, we can deduce from Fig. 9 that the ResNet50 CNN (and possibly CNNs in general) do indeed encode and maintain information about the number of objects contained within an image. From an information-theoretic perspective, it was important to know this because if information was lost then there would have been little plausibility in attempting to repurpose CNNs as counting networks.

## 6 CONCLUSION

In the previous section we presented the different results we attained for our three experiments. We learned many insights which we would like to summarize below.

First, the problem of object detection can be reformed into a regression problem. We first unsuccessfully tried to regress directly on bounding boxes. By using Soft Attention instead we were able to isolate some objects from the background. The approach needs a closer investigation to understand the dynamics involved in the process of selection and isolation of objects.

We recognize the current limitation that the network was able to isolate only one object. We perceive this can be tackled by 'subtracting' detected objects from the original image and reiterating over the image.

Second, the Object Location network has trained well, achieving a high validation accuracy in its classification task. It tells us that CNNs can indeed be used to provide qualitative labels for an object's location.

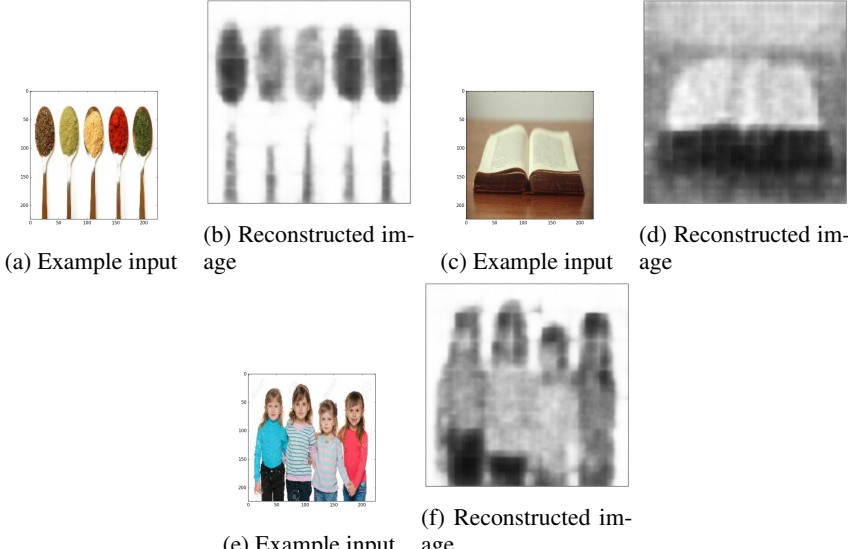

(a) Example input

(b) Reconstructed image

(c) Example input

(d) Reconstructed image

(e) Example input

(f) Reconstructed image

Figure 9: The figure shows sample inputs to the Plurality Autoencoder net and their respective reconstructions. We can see that the network creates the appropriate number of 'blobs' per each image

We recognize the limitation of only using images with a single object against a plain background, as the real-world environment is cluttered. We perceive this can be tackled by creating more specialized CNNs for feature extraction that would be immune to noisy backgrounds.

Third, we found that CNNs such as ResNet50 do indeed perceive and encode object plurality. We found that this information can be recovered as we demonstrated in Fig. 9. To this end, it is thus plausible to further investigate why it was difficult for us to reframe counting objects as a classification problem.

To conclude, we presented three thrusts in weakly and semi-supervised computer vision. The purpose of our experiments was to demonstrate the plausibility of tackling these problems in the way we do.

Our work is not concerned with state-of-the-art accuracy or performance. An open-ended child robot does not strictly need such capacities. What we are concerned with, however, is the potential for self-improvement under weakly or unsupervised conditions. For example, by reframing detecting an object's qualitative location into a classification problem, we can acquire a label from a human teacher in natural language, and use it to correct the robot's asseessment of the location of an object in its field of view. In such a way, we believe robots can scale their learnings and improve by consolidating knowledge over time.

# 7 REFERENCES

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
