# OpenReview forum: "TOWARDS ROBOT VISION MODULE DEVELOPMENT WITH EXPERIENTIAL ROBOT LEARNING"
_ICLR.cc/2018/Conference — Reject_

### Official Review · AnonReviewer1 · 2017-11-26
**Nothing new**

**Rating:** 3
**Confidence:** 4

**Review:**

The paper is motivated with building robots that learn in an open-ended way, which is really interesting. What it actually investigates is the performance of existing image classifiers and object detectors. I could not find any technical contribution or something sufficiently mature and interesting for presenting in ICLR.

Some issues:
- submission is supposed to be double blind but authors reveal their identity at the start of section 2.1.
- implementation details all over the place (section 3. is called "Implementation", but at that point no concrete idea has been proposed, so it seems too early for talking about tensorflow and keras).

---

### Official Review · AnonReviewer3 · 2017-11-27
**Random ideas about object detection with poor discussion**

**Rating:** 2
**Confidence:** 3

**Review:**

This work explores some approaches in the object detection field of computer vision: (a) a soft attention map based on the activations on convolutional layers, (b) a classification regarding the location of an object in a 3x3 grid over the image, (c) an autoencoder that the authors claim to be aware of the multiple object instances in the image. These three proposals are presented in a framework of a robot vision module, although neither the experiments nor the dataset correspond to this domain.

From my perspective, the work is very immature and seems away from current state of the art on object detection, both in the vocabulary, performance or challenges. The proposed techniques are assessed in a dataset which is not described and whose results are not compared with any other technique. This important flaw in the evaluation prevents any fair comparison with the state of the art.

The text is also difficult to follow. The three contributions seem disconnected and could have been presented in separate works with a more deeper discussion. In particular, I have serious problems understanding:

1. What is exactly the contribution of the CNN pre-trained with IMageNet when learning the soft-attention maps ? The reference to a GAN architecture seems very forced and out of the scope.

2. What is the interest of the localization network ? The task it addresses seems very simple and in any case it requires a manual annotation of a dataset of objects in each of the predefined locations in the 3x3 grid.

3. The authors talk about an autoencoder architecture, but also on a classification network where the labels correspond to the object count. I could not undertstand what is exactly assessed in this section.

Finally, the authors violate the double-bind review policy by clearly referring to their previous work on Experiental Robot Learning.

I would encourage the authors to focus in one of the research lines they point in the paper and go deeper into it, with a clear understanding of the state of the art and the specific challenges these state of the art techniques may encounter in the case of robotic vision.

---

### Official Review · AnonReviewer2 · 2017-11-29
**Paper violates double bind**

**Rating:** 2
**Confidence:** 4

**Review:**

The authors disclosed their identity and violated the terms of double blind reviews.
Page 2 "In our previous work (Aly & Dugan, 2017)

Also the page 1 is full of typos and hard to read.

---

### Decision · Program_Chairs · 2018-01-29
**ICLR 2018 Conference Acceptance Decision**

**Decision:**

Reject

**Comment:**

Reviewers unanimous on rejection.
Authors don't maintain anonymity.
No rebuttal from authors.
Poorly written